# Recycling polyolefin plastic waste at short contact times via rapid joule heating

Esun Selvam [1,2,4], Kewei Yu [2,4], Jacqueline Ngu[1,2], Sean Najmi [2,3] & Dionisios G. Vlachos [1,2,3] ✉

The chemical deconstruction of polyolefins to fuels, lubricants, and waxes offers a promising strategy for mitigating their accumulation in landfills and the environment. Yet, achieving true recyclability of polyolefins into $C_2$-$C_4$ monomers with high yields, low energy demand, and low carbon dioxide emissions under realistic polymer-to-catalyst ratios remains elusive. Here, we demonstrate a single-step electrified approach utilizing Rapid Joule Heating over an H-ZSM-5 catalyst to efficiently deconstruct polyolefin plastic waste into light olefins ($C_2$-$C_4$) in milliseconds, with high productivity at much higher polymer-to-catalyst ratio than prior work. The catalyst is essential in producing a narrow distribution of light olefins. Pulsed operation and steam co-feeding enable highly selective deconstruction (product fraction of >90% towards $C_2$-$C_4$ hydrocarbons) with minimal catalyst deactivation compared to Continuous Joule Heating. This laboratory-scale approach demonstrates effective deconstruction of real-life waste materials, resilience to additives and impurities, and versatility for circular polyolefin plastic waste management.

Polyolefins, comprising low-density and high-density polyethylene (LDPE, HDPE) and polypropylene (PP), constitute ~60% of the plastic waste[1]. While mechanical recycling is pivotal in managing some plastic waste, it falls short in producing high-quality materials, especially from polyolefins and mixed plastics[2,3]. Consequently, there has been a surge in chemical upcycling and recycling technologies. However, activating C-C bonds in polyolefins is challenging, rendering these processes energy-intensive and unselective to high-value products. Advancements in hydrocracking[4,5], hydrogenolysis[6–8], and catalytic pyrolysis[9,10] have produced fuels, lubricants, and waxes with relatively high selectivity. Nevertheless, these processes are not directly circular since their products need additional steps to produce monomers. An approach to synthesizing monomers can entail producing naphtha via hydrocracking followed by steam cracking at high temperatures[5] (Pathway 1 in Fig. 1). However, this two-step process is very energy intensive. Alternatively, catalytic pyrolysis[10,11] and ethenolysis[12,13] have also been explored for direct monomer production. Ethenolysis employs ethylene as a reagent, the monomer of polyethylene, along with expensive

noble metal catalysts[12,13], making it less circular and economically viable. Catalytic pyrolysis, while promising, entails high catalyst-to-polymer ratios (»1), lower selectivity to ethylene and propylene, and high energy intensity due to elevated reaction temperatures (especially when compared to mechanical recycling)[14] along with slow penetration of the heat to the plastic.

Process electrification can play a crucial role in decarbonizing the chemical industry by enabling the transition from traditional fossil fuel-based processes to more sustainable and energy-efficient alternatives[15,16]. Utilizing renewable electricity as a clean energy source can replace carbon-intensive methods in producing key chemicals[16]. As a result, electrified catalytic and non-catalytic processes using microwaves[17–19], induction heating[20,21], and Joule (or resistive) heating[22,23] have gained interest. Recent work by Dong et al. demonstrated the potential of spatiotemporal heating (STH), a form of Joule heating, in achieving selective monomer production from plastics, even in the absence of a catalyst[23]. Furthermore, Mastalski et al. illustrated that thin films can eliminate heat and mass transfer

[1]Center for Plastics Innovation, University of Delaware, 221 Academy St., Newark, DE, USA. [2]Department of Chemical and Biomolecular Engineering, University of Delaware, 150 Academy St, Newark, DE, USA. [3]Delaware Energy Institute, University of Delaware, 221 Academy St., Newark, DE, USA. [4]These authors contributed equally: Esun Selvam, Kewei Yu. ✉e-mail: vlachos@udel.edu

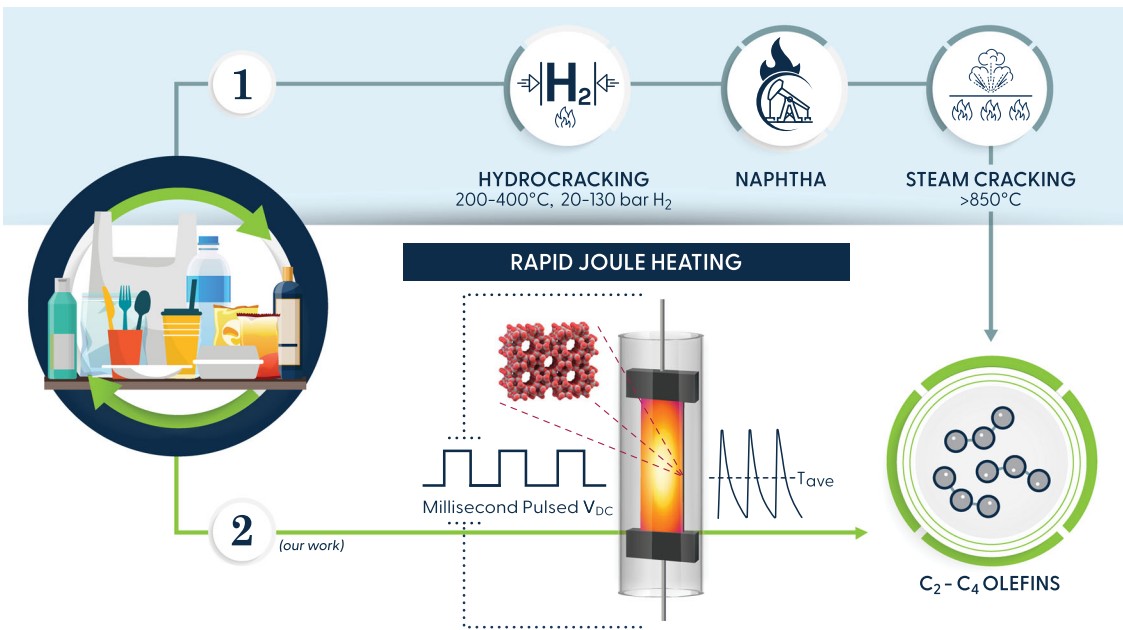

**Fig. 1 | Chemical pathways of converting polyolefins to monomers.** Path 1 involves plastics hydrocracking at high hydrogen pressures to produce naphtha, which is then converted to $C_2$–$C_4$ olefins via steam cracking at high temperatures. Path 2 (this work) demonstrates the direct conversion of polyolefins via rapid pulse or continuous Joule heating over H-ZSM-5 catalysts.

limitations[24]. However, limited work has been conducted on catalytic deconstruction.

This work presents a laboratory-scale approach to selectively deconstruct polyolefins into $C_2$–$C_4$ olefins using Rapid Pulse Joule Heating (RPH) over an H-ZSM-5 catalyst (Pathway 2 in Fig. 1). We investigate the influence of different reactor parameters on polymer conversion and product distributions. We also compare the efficacy of RPH to Continuous Joule Heating (CJH) in deconstructing waste and catalyst deactivation. Additionally, we demonstrate that co-feeding steam enhances the yield of light olefins and reduces coking. Finally, we showcase the potential of this approach to effectively address various real-world plastic wastes.

## Results
### Optimization of reaction parameters
The depolymerization of the plastic wastes was conducted in a Joule Heated reactor (Supplementary Figs. 2–3) consisting of a carbon fiber paper (CFP) impregnated with an MFI zeolite (H-ZSM-5) and coated with the plastic. The polymer is in intimate contact with the CFP and the catalyst and is heated resistively very rapidly by the heating element. The flowing He gas, regulated by a mass flow controller, entrains the gas products and minimizes their contact with the catalyst and secondary reactions. Reaction conditions were optimized to maximize conversion and product fractions of monomers (ethylene, propylene, and potentially butylene) under pulse heating. First, the DC voltage was adjusted to control the average and peak temperatures. The total pulsing time was 60 s, encompassing 60 pulses, with each pulse comprising 50 ms of heating and 950 ms of cooling and a polymer-to-catalyst ratio of ~1. Non-catalytic tests were also performed as a reference. The excellent electric and thermal conductivity of the CFP enables rapid and uniform heating, respectively. Figure 2a demonstrates that the catalyst significantly increases (~20% higher) activity at 32 V ($T_{max} = 540\ ^\circ C$) but gives comparable conversion at 42 V because of higher peak temperatures ($T_{max} = 750\ ^\circ C$). However, the product distributions differ profoundly: non-catalytic cracking leads to a broad distribution of alkanes and olefins ranging from $C_1$–$C_{31}$ along with heavier wax products (Supplementary Figs. 5–6), whereas catalytic cracking results in complete conversion and produces $C_1$–$C_8$ range hydrocarbons, with a product fraction of ~74% for

$C_2$–$C_4$ products. A voltage of 42 V achieved the highest product fraction of $C_2$–$C_3$ products (~45%) and was selected for subsequent tests. We attribute the high product fractions of light olefins to the thin polymer film (< 20 μm thick) and short contact times. This bears critical importance, as a thin polymer film (e.g., <100 μm at ~700 °C) ensures negligible heat and mass transport limitations ($t_{reaction} > t_{convection} > t_{conduction}$; $t_{reaction} > t_{diffusion}$), and short contact times are essential to preclude slower secondary reactions leading to char formation[24]. Furthermore, this finding also corroborates prior literature suggesting that short residence times can suppress aromatics formation and enhance selectivity towards olefins for catalytic pyrolysis of polyolefins[25,26].

Once the voltage was optimized, we studied the effect of the number of pulses on conversion (Fig. 2b). ~10 pulses achieved full conversion, whereas ≤3 pulses exhibited significantly lower conversion. The lower conversion is attributed to lower peak and average temperatures ($T_{max} = 503\ ^\circ C$ at 3 pulses vs. 730 °C at ≥5 pulses), resulting from the energy of the initial pulses being expended in heating the polymer to its melting point and facilitating the phase transition from solid to molten state. Additionally, the lower peak and average temperatures result in lower fractions of $C_2$–$C_4$ hydrocarbons. The effect of the polymer-to-catalyst ratio was studied next. All ratios (1.25–20) show >90% LDPE conversion (Fig. 2c), but the amount of coke increases monotonically with increasing ratio, as depicted in Fig. 2d and Supplementary Fig. 6. Raman spectra of the spent catalyst indicated an intense graphitic band at ~1610 cm⁻¹, along with two defected carbon bands at ~1395 cm⁻¹ and ~1210 cm⁻¹, respectively[27]. At catalyst-to-polymer ratios of <0.4, heavy coke exceeds 30 wt. % (Supplementary Fig. 7). Hence, a ratio of ~0.8 (polymer-to-catalyst ratio ~1.25) was used for subsequent tests. Lastly, varying the He gas flow rate led to no significant changes (Fig. 2e).

### Effect of pulsing parameters
We also examined the effect of pulsing frequency through systematic variation of heating and cooling times. First, we modulated the frequency by varying the cooling time while keeping the heating time at 50 ms and maintaining a voltage of 42 V for 5 pulses. Figure 3a illustrates notably lower conversions at 0.25 Hz and 0.5 Hz, attributed to the longer cooling times (3950 ms and 1950 ms, respectively) that lead

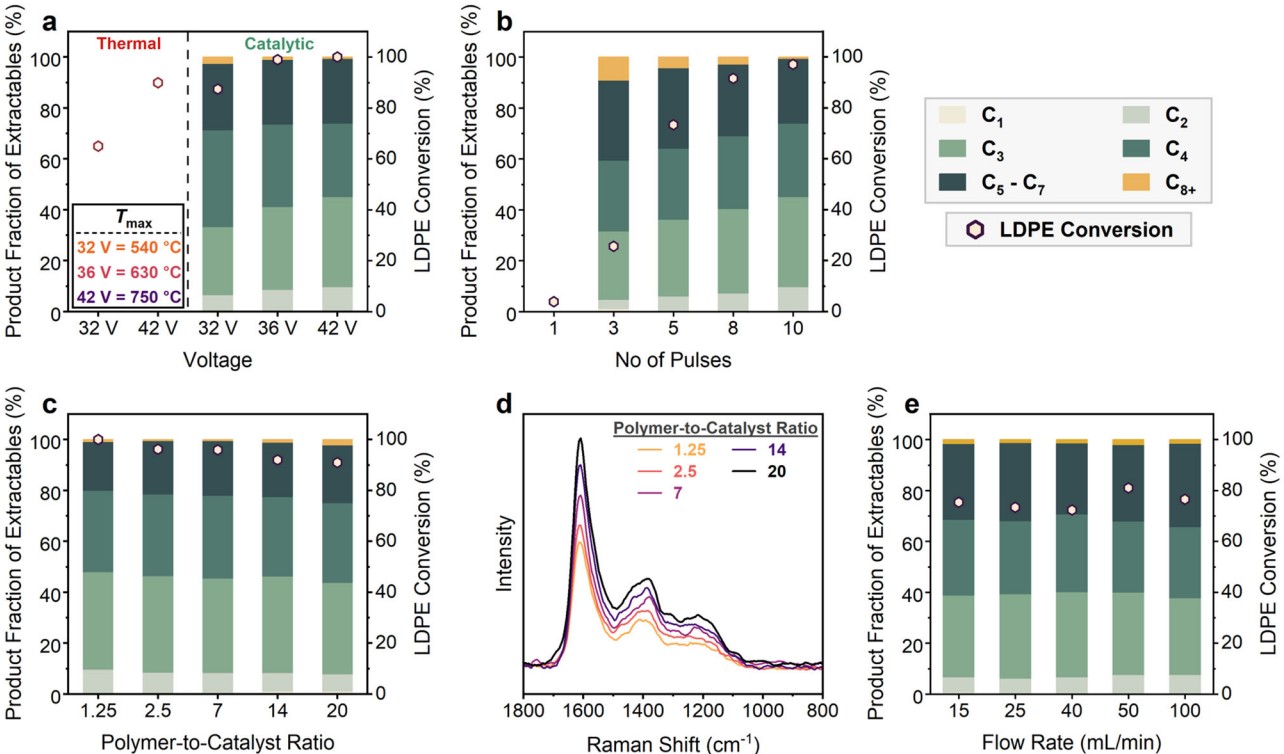

**Fig. 2 | Performance of thermal and catalytic rapid pulse heating (RPH) on LDPE deconstruction at various operating conditions. a** Effect of DC voltage on the performance of thermal and catalytic RPH (60 pulses). **b** Effect of the number of pulses on the performance of RPH of LDPE over H-ZSM-5 catalyst (42 V). **c** Effect of polymer-to-catalyst ratio on LDPE conversion (42 V and 10 pulses). **d** Raman spectra of spent catalysts demonstrating increased coking at higher polymer-to-catalyst ratios. (e) Effect of the He gas flow rate on the performance of RPH of LDPE over H-ZSM-5 catalyst (42 V and 5 pulses).

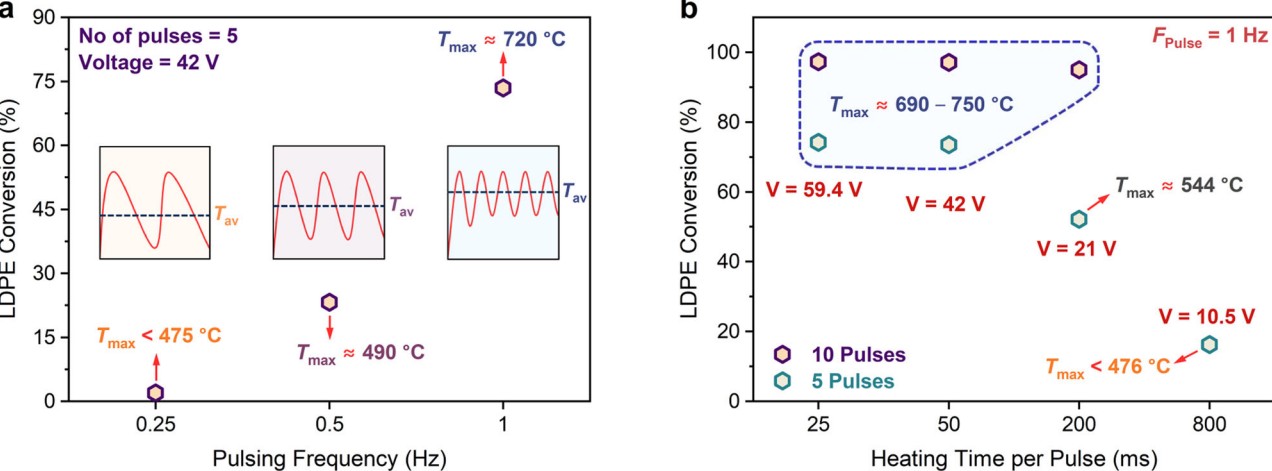

**Fig. 3 | Pulsing effect on conversion.** Effect of (**a**) pulsing frequency (controlled by varying cooling times) and (**b**) heating time on the performance of rapid pulse heating (RPH) for LDPE deconstruction at constant energy consumption. The insets in (**a**) depict the pulses.

to lower peak and average temperatures (Fig. 3a, Supplementary Table 1). For example, at 0.25 Hz, the cooling time is ~4 s, causing the CFP to reach near room temperature upon cooling. Likewise, the conversion after two sequential experiments of 5 pulses each on the same sample at 1 Hz is ~20% lower than in a single sequence of 10 pulses (Supplementary Fig. 8). Similarly, at a given frequency, fewer pulses reduce the conversion significantly (Fig. 2b). A similar phenomenon was reported by Dong et al., demonstrating that the non-catalytic process requires a combination of melting, wicking, vaporization, and reaction to achieve good performance[23]. However, with

rapid joule heating alone without a bilayer structure, the non-catalytic depolymerization is unselective due to the lack of a temperature gradient that promotes wicking. Additionally, we assessed the effect of pulsing frequency whilst maintaining the peak temperatures ($T_{max}$ ~ 720 °C) and the total exposure time (8 seconds) constant. This was accomplished by adjusting the DC voltage and pulse sequences: 2 pulses at 0.25 Hz (65 V), 4 pulses at 0.5 Hz (53 V), and 8 pulses at 1 Hz (42 V). Again, the results suggest that the conversions decrease with a decrease in frequency, primarily due to a lower $T_{avg}$ (Supplementary Fig. 9).

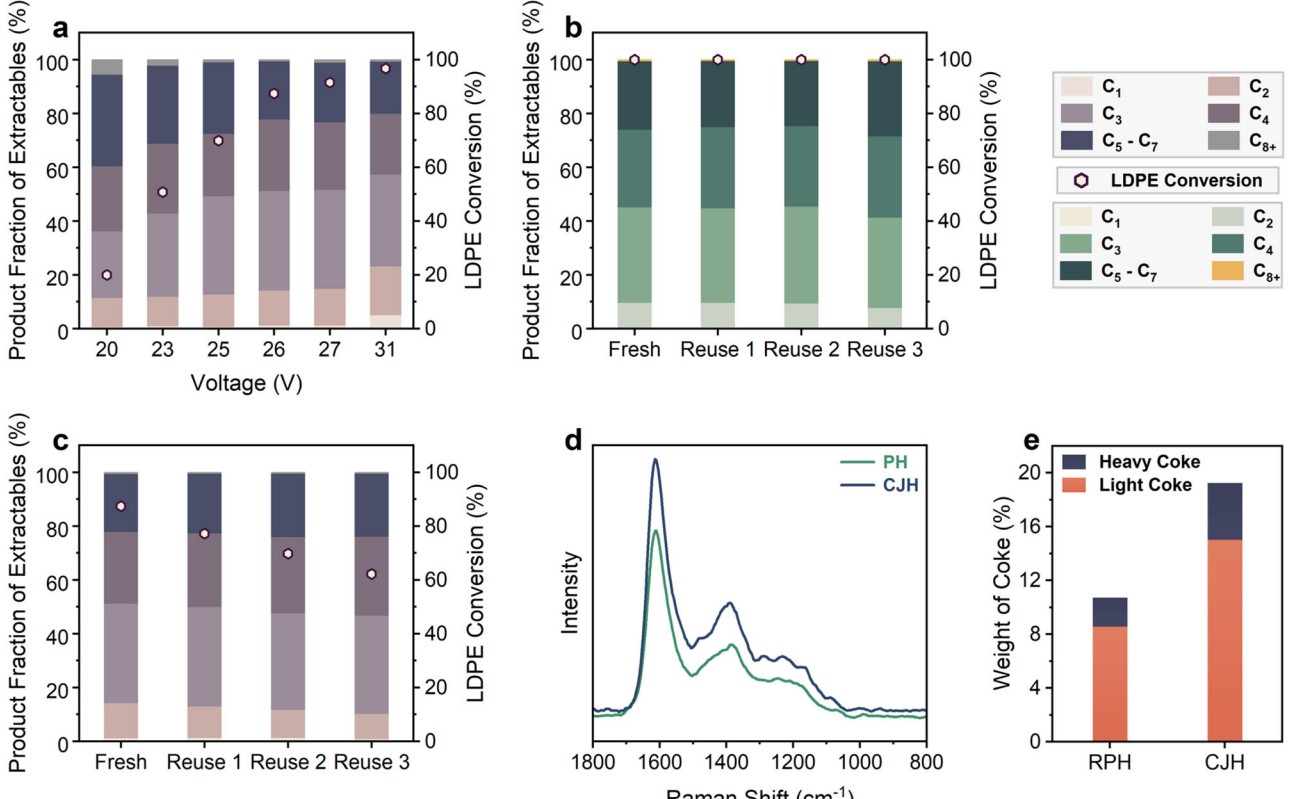

**Fig. 4 | Heating mode performance and catalyst reuse comparison. a** Effect of DC voltage on conversion and product fractions of extractable products for CJH of LDPE over H-ZSM-5 (reaction time = 500 ms). Reusability of CFP coated with H-ZSM-5 for (**b**) pulse heating (42 V and 10 pulses) and (**c**) CJH of LDPE (26 V and

500 ms). **d** Raman spectra of the spent H-ZSM-5 catalyst after 3 cycles of reuse for RPH and CJH of LDPE. **e** Weight % of coke obtained from TGA of spent catalysts after 4 reuse cycles for RPH and CJH of LDPE.

Alternatively, we investigated variations in the pulse width while maintaining a constant pulsing frequency of 1 Hz by adjusting the heating and cooling times. In these experiments, the energy consumption was held nearly constant by regulating the voltage and pulse width simultaneously (for example, a 50 ms pulse width at 42 V and a 200 ms pulse width at 21 V consume the same amount of energy due to the equation, $E = t*U^2/R$, where $t$ is the pulse width, $U$ is the voltage, and $R$ is the resistance of the CFP). 10 pulses gave no distinct change in conversion and product fractions, likely due to a similar $T_{max}$ (~690–750 °C) (Fig. 3b). 5 pulses gave much lower conversion with heating times exceeding 200 ms, primarily due to significantly lower temperatures ($T_{max}$ = 544 °C at 21 V and $T_{max}$ < 476 °C at 10.5 V) (Fig. 3b). Similar trends were seen for product fractions, with an increase in heavier products at lower temperatures. The average and peak temperatures are key process parameters, with high temperatures being essential.

### Continuous Joule Heating (CJH) vs. Rapid Pulse Joule Heating (RPH)

Next, we compared CJH as an alternative heating strategy. To ensure a fair comparison, the DC power source was kept operational for 500 ms, equivalent to the total heating time (voltage on) for 10 pulses in RPH. The voltage was varied between 20–31 V, affecting $T_{avg}$. An increase in the voltage increased conversion, and a voltage of 31 V achieved full conversion (Fig. 4a). The fraction of $C_2$–$C_3$ products also increased with voltage, plateauing around 26–27 V (Fig. 4a). Further increasing the voltage enhanced the yields of lighter products, including methane, acetylene, and propyne. Our results demonstrated that CJH exhibits slightly higher product fractions of $C_2$–$C_4$ products compared to RPH, primarily due to the elevated $T_{avg}$ at similar $T_{max}$ (Supplementary Table 1). A voltage of 26 V ($T_{max}$ ~ 730–750 °C) closely

resembled our optimized RPH conditions and was chosen for subsequent CJH experiments.

While CJH slightly increases the fraction of light olefins produced compared to RPH, pulsed operation often offers other co-benefits, for instance, it can reduce coke due to shorter exposure at elevated temperatures[28,29]. Thus, we investigated the reusability of the CFP in RPH and CJH modes of operation. Prior to each cycle, the CFP was coated with polymer following the procedure described in the Methods section without washing or regenerating the catalyst. Figure 4b illustrates that, as hypothesized, the LDPE conversion and product distributions remain unchanged in 3 cycles of reuse in RPH. In contrast, CJH exhibits a decline in conversions and fraction of light olefins produced upon reuse (Fig. 4c). We attribute this diminishing activity to higher coking in CJH than the RPH mode. It is widely accepted that hydrocarbon reactions at higher temperatures over solid acids coke more due to alkylation, cyclization, and dehydrogenation[30–32]. The higher $T_{avg}$ in CJH (~625 °C) leads to expedited catalyst deactivation than RPH (~540 °C). Raman spectra (Fig. 4d) corroborate significantly higher coking in CJH, demonstrating an $I_D/I_G$ ratio (the ratio of the Defected to Graphitic Carbon) of 0.44 (vs. 0.38 for RPH). Consistently, TGA data (Fig. 4e and Supplementary Fig. 10) shows nearly twice the amount of coke, with more heavy coke, ultimately rendering its performance inferior.

### Effect of co-feeding steam

It is known that co-feeding steam during naphtha/hydrocarbon cracking enhances the selectivity of light olefins[33]. Likewise, the efficacy of steam in mitigating coking has also been previously documented in several reactions[34,35]. Hence, we investigated the impact of co-feeding steam. The He gas was introduced through a water-filled bubbler in these tests, with the rest of the experimental setup

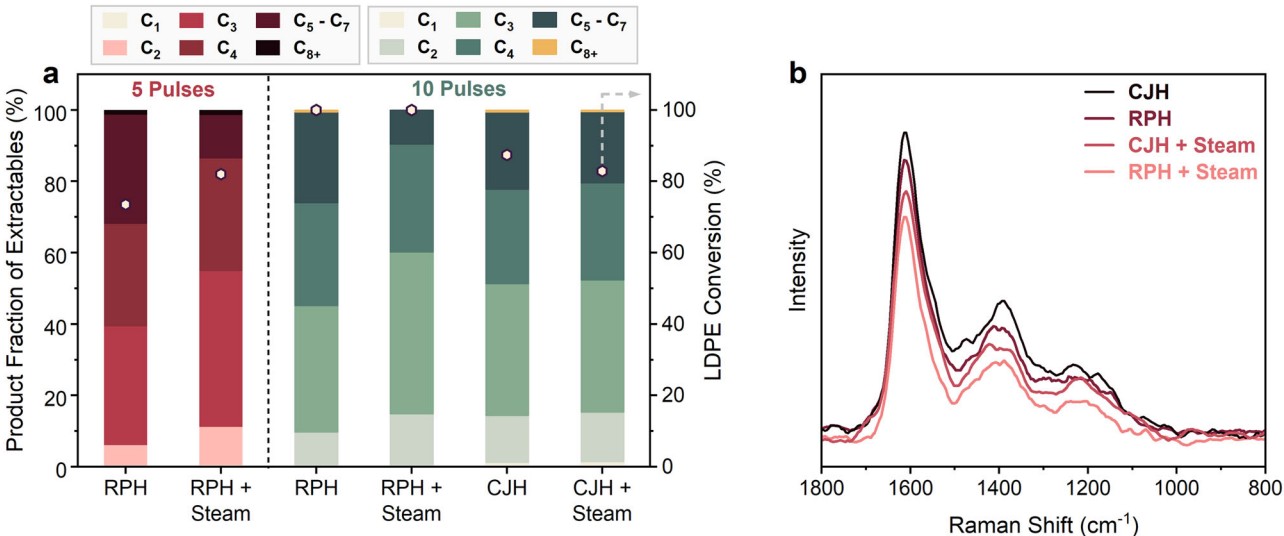

**Fig. 5 | Effect of Co-feeding Steam on Performance. a** Effect of co-feeding steam on RPH of LDPE over H-ZSM-5 catalyst (42 V, $T_{max}$ = 730 °C). **b** Raman spectra of spent catalysts for RPH and CJH.

unchanged. The results (Fig. 5a) demonstrate a substantial enhancement in the fractions of $C_2$–$C_4$ olefins when steam was co-fed in RPH (increases by ~16% over RPH), surpassing the performance of even the CJH configuration discussed above. No significant change in the conversion was observed. The combined yield to $C_2$–$C_4$ products is ~90 wt%. Similarly, we also observed a slight increase in the $C_2$–$C_4$ olefin fractions when steam was co-fed in CJH. However, this increase is not as pronounced as in RPH, underscoring the importance of high-temperature pulsing. As hypothesized, the enhanced performance is likely attributable to reduced coking, consistent with prior work[34,35]. Raman spectra of the spent catalysts further illustrate the lowest amount of coke when steam was co-fed (Fig. 5b). The $I_D/I_G$ ratio was the lowest for RPH with steam ($I_D/I_G$ = 0.358) followed by CJH with steam ($I_D/I_G$ = 0.39) compared to RPH ($I_D/I_G$ = 0.396) and CJH ($I_D/I_G$ = 0.44). However, in CJH we see still observe catalyst deactivation with steam co-feeding after the 2$^{nd}$ cycle of reuse (Supplementary Fig. 11, $I_D/I_G$ – RPH = 0.4 < $I_D/I_G$ – CJH with Steam = 0.42 < $I_D/I_G$ – CJH = 0.44, after 3 cycles of reuse). TGA further demonstrates the least coke (~10 wt%) when steam was co-fed. Further analysis revealed that co-feeding steam reduces the amount of heavy coke (~16% with steam vs. ~20% in RPH and ~22% in CJH). Differential Thermal Analysis (DTA) further illustrates significant changes in the maxima positions among samples, with the peaks shifting to lower temperatures by ~5–15 °C when steam was co-fed (Supplementary Fig. 12).

### Performance of RPH with various feedstocks and real-world plastics

Real-world plastics contain additives and impurities that can hinder the catalyst efficacy[36–38]. Hence, we investigated the deconstruction of PE and PP feedstocks from diverse sources. Figure 6a illustrates that the conversions and product distributions remain largely unaffected, irrespective of source, highlighting the process's resilience. Notably, PP materials yield a higher fraction of $C_2$–$C_4$ range products, likely stemming from the easier activation of the tertiary carbons and faster β-scission rates of tertiary carbenium ion intermediates[39,40]. In contrast, PE, predominantly composed of secondary carbons, necessitates isomerization before undergoing cracking[19]. Strikingly, dyes in fishnets (cyan) and centrifuge tubes (black gradations) do not impact performance. The plastic grocery bags made of HDPE achieve a slightly lower 'apparent conversion' of ~84%, possibly due to calcium carbonate ($CaCO_3$), extensively used as a filler in plastic bags. TGA of the plastic bags indicated that $CaCO_3$ constitutes roughly 20 wt% (Supplementary

Fig. 15). During the reaction, $CaCO_3$ partly decomposes to CaO, which accounts for the remaining unconverted mass, i.e., the filler does not seem to impact the polymer conversion.

## Discussion

The proposed Rapid Pulse Joule Heating (RPH) for catalytic cracking of polyolefin waste exhibits significant promise in depolymerizing polyolefins to monomers. The ultra-fast reactor heating, enabled by the CFP, allows plastic deconstruction in 500 ms (via 10, 50 ms pulses). H-ZSM-5 catalyst enables the effective breakdown of polymers into light hydrocarbons, with a $C_2$–$C_4$ product fraction of >75% at full conversion. Furthermore, this technology surpasses state-of-the-art studies in the literature, achieving high productivity at ~50–200 times lower catalyst usage, as illustrated in Fig. 6c and Supplementary Fig. 22. While continuous Joule heating (CJH) offers improved light olefin production compared to RPH, it exhibits significantly more catalyst deactivation. Co-feeding steam in RPH showcases increased monomer production by minimizing coke formation, outperforming CJH. Notably, we demonstrate a product fraction of >90% towards $C_2$–$C_4$ hydrocarbons at full conversion, highlighting the RPH's potential for monomer production. The high propylene-to-ethylene ratios (~2.6–3.5) suggest that the technology is particularly well suited for propylene production. Compared to conventional catalytic pyrolysis, the thin-film structure of the polymer eliminates heat transfer limitations and the associated bulky reactors to enable modular systems. Furthermore, RPH's efficacy with various real-life feedstocks opens new avenues for monomer production from polyolefins, presenting a sustainable approach while mitigating associated $CO_2$ emissions. While the proposed electrified reactor demonstrates promise for monomer production from plastic waste, it is currently a prototypical laboratory-scale framework with limited potential for commercialization. Further development is necessary, particularly in reactor design to facilitate scale-up and continuous operation, as well as in optimizing product recovery methods. Additionally, benchmarking at larger scales is essential to facilitate practical large-scale implementation.

## Methods

### Materials

LDPE ($M_W$ = 4 kDa), PP ($M_W$ = 12 kDa), ethanol (EtOH), toluene, xylene, and dichloromethane (DCM) were obtained from Sigma-Aldrich. H-ZSM-5 (CBV3024E, Si:Al = 30, $NH_4^+$-form) was obtained from Zeolyst International. Carbon fiber paper (CFP) - Freudenberg H23, thickness =

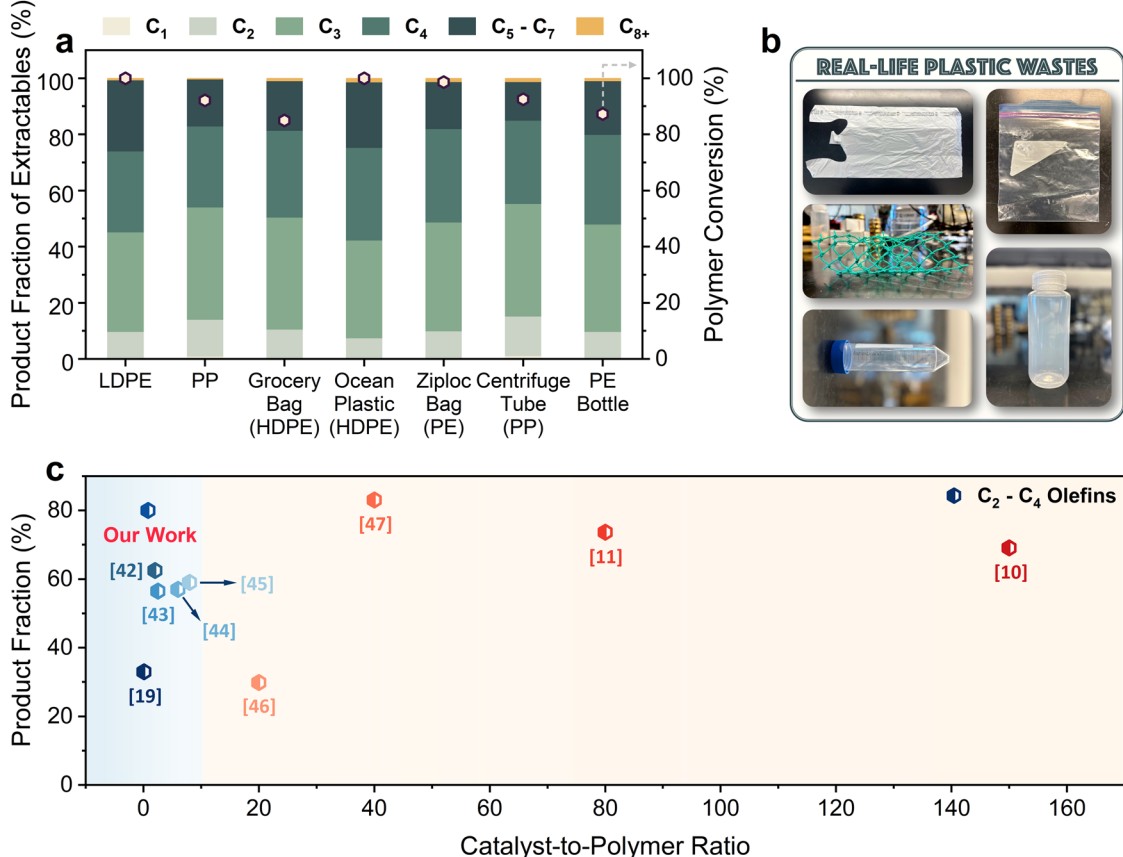

**Fig. 6 | Performance for various feedstocks and of other literature studies.**
**a** Performance of rapid pulse heating (RPH) for the deconstruction of real-life plastics over H-ZSM-5 catalyst (42 V, 10 pulses and $T_{max}$ = 730 °C). **b** Images of real-life grocery bags, fishnets, centrifuge tubes, Ziploc™ bags, and PE bottles were used in this study. **c** Comparison of the product fractions of $C_2$–$C_4$ olefins vs. catalyst-to-polymer ratio of this work to literature for the catalytic deconstruction of PE[10,11,19,42–47]. Results are at high ( - 100%) conversion.

210 μm, was obtained from Freudenberg Performance Materials SE & Co. Polyethylene bottles were obtained from SP Scienceware. PE bags were obtained from Ziploc™. HDPE grocery bags were obtained from ULINE. Centrifuge tubes were obtained from Fisher Scientific. The Ocean plastic (HDPE fishnets) was provided by researchers at the Hawaii Pacific University.

## Heating element preparation

Catalysts were coated on a CFP heating element with an exposed area of 38 mm by 8 mm (Supplementary Fig. 1). Prior to coating, the ZSM-5 was calcined in air at 550 °C (2 °C/min ramp) for 4 h to convert it to H⁺ form (referred to as H-ZSM-5 henceforth). Typically, 10 mg of catalyst was sonicated in EtOH to form a uniform suspension before being drip-coated uniformly on both sides of the CFP (Supplementary Fig. 1b). The suspension was self-distributed on the CFP by the capillary effect. Once the catalyst was coated, the CFP was dried in an oven at 70 °C in static air to remove excess solvent.

A given amount of polymer and toluene (or xylene in the case of HDPE and PP) were mixed and heated to 130–180 °C to obtain a homogeneous solution of the polymer in the solvent (at a concentration of 5-20 wt%). Once the polymer was completely dissolved, the CFP was placed on a glass slide on a hotplate at -150 °C. The polymer solution was then added dropwise to the CFP and left on the hot plate for 2 min to remove residual solvent. Any excess solvent was removed by further drying the CFP in a vacuum for 30 min at 70 °C. Supplementary Fig. 1 shows SEM images of uniformly coated polymer on the CFP with H-ZSM-5. The bright regions (Supplementary Fig. 1c) represent the polymer's negative charging during SEM imaging[41]. Given the porous structure of the CFP matrix, the polymer deposits on the fibers and within the inter-fiber gaps. SEM images (Supplementary Fig. 1) reveal that the distance between adjacent fibers is typically <20 μm. Consequently, the thickness of the polymer layer is expected to be <20 μm.

## Joule Heated (JH) Reactor Design

The JH reactor is a vertical quartz tube (ID = 22 mm) with gas reactants flowing from top to bottom (Supplementary Figs. 2–3). The Joule heating element is fixed by two stainless steel clamps and four copper plates to ensure good electrical contact (Supplementary Fig. 3). A direct current (DC) power supply provides current for Joule heating (Volteq HY7520ex). A DC solid-state relay, a source meter (Keithley 2425), and a LabView program enable programmable heating. The temperature of the heating element is monitored using an IR camera (Optris PI 1 M or PI640) through the quartz reactor with a frame rate of up to 1000 FPS (frames per second) (Supplementary Fig. 3). The reactor is subjected to passive air cooling (20 °C) in a fume hood. A simplified schematic of the setup is shown in Supplementary Fig. 2. A typical RPH has a cycle period of 1 s, with 5% of duty cycle (50 ms heating and 950 ms cooling). The pulsing parameters are adjusted accordingly as per experimental requirements. Alternatively, the heating element can operate under a steady state (CJH).

## Reactivity Tests

The conversion of the plastic wastes was conducted in the JH reactor (Supplementary Figs. 2–3) consisting of a CFP impregnated with the zeolite coated with -10–12 mg of plastic along with an upward He flow of 25 sccm (unless stated otherwise) regulated by a mass flow controller. The He gas flowing along CFP acts as a carrier gas and removes the vaporized product. A Tedlar bag was connected to the rear end of

the reactor section for online collection of the hydrocarbons produced. For steam co-feeding experiments, the He gas was bubbled through a bubbler containing water, resulting in a concentration of ~3-4% water in the gas stream.

## Catalyst characterization

The X-ray diffraction (XRD) pattern of the catalyst was recorded using an X-ray diffractometer (Bruker D8) with Cu Kα radiation ($\lambda = 1.54056$ Å) at 40 kV and 40 mA and a scanning rate of 0.05 per second between $2\theta = 10-70°$. Elemental composition was obtained using X-ray fluorescence (XRF) spectroscopy on a Rigaku WDXRF. $N_2$ physisorption at $-196\,°C$ was performed on a Micromeritics ASAP 2020 instrument. Fourier transform infrared (FTIR) spectra of adsorbed pyridine followed by pyridine thermodesorption were recorded in transmission mode in a homemade Pyrex tubular flow cell equipped with 32 mm KBr windows. The sample was pressed in a self-supported wafer ( ~15 mg, 1.3 cm² and 40 bar/cm² pressure), placed in a quartz sample holder, and heated in the flow of pure Ar at 300 °C (ramping rate 10 °C/min) with 1 h dwell at that temperature. Then, the temperature was reduced to 150 °C, and the sample was treated with pyridine vapor by injecting liquid pyridine (5 µl, 99.8%; Sigma-Aldrich) with a micro syringe through a septum port. After saturation, the sample was flushed with pure He for 30 min, and the spectrum of the pyridine-saturated sample was recorded. Finally, the temperature was increased from 10 °C/min to 300 °C in a constant flow of Ar, and spectra were recorded every 1 min. Integration and peak deconvolution were done using the OMNIC 8.2 software. The coked catalyst samples were collected by sonicating the spent CFP in EtOH, followed by drying the samples in the air at 70 °C. Thermogravimetric analysis was performed on a TGA5500 (TA instrument). ~5 mg of sample was loaded onto each pan and heated in 20 sccm of air at a ramp rate of 20 °C/min. Raman spectra of the coked catalyst samples were recorded under ambient conditions on a Horiba LabRam microscope with a 15x objective using a 325 nm UV laser. The analysis of the spectra was done using OMNIC 8.2 software.

## Product analysis

The products collected in the Tedlar gas sampling bag were analyzed with a GC-FID (Agilent HP-PLOT/Q GC column). Additionally, a solvent trap filled with $CH_2Cl_2$ and cooled down to ~0 °C was used to extract products with $\geq C_4$, and the species in the resulting mixture were analyzed using a GC–MS (Agilent DB-5 column; Supplementary Fig. 17). Calibration coefficients and retention times for all products were measured using $C_1-C_8$ analytical standards (Supplementary Figs. 18–21). The conversion of LDPE was calculated according to Eq. (1):

$$\text{LDPE/Polymer Conversion}(\%) = \frac{W_{\text{CFP,i}} - W_{\text{CFP,f}}}{W_{POL,i}} \times 100\% \qquad (1)$$

where $W_{\text{CFP,i}}$ is the initial weight of the CFP, after coated with the polymer and the catalyst, $W_{\text{CFP,f}}$ is the final weight of the CFP, and $W_{\text{POL,i}}$ is the weight of the polymer coated calculated by measuring the weight difference of CFP before and after coating the polymer.

It must be noted that the calculated conversion here does not consider the inevitable loss of polymer into coke (typically <10% based on TGA). Hence, the real conversion is expected to be slightly higher than the reported conversion.

The product fraction of extractable products (gas and liquids) was calculated according to Eq. (2):

$$\text{Product fraction of Extractables}(\%) = \frac{\text{Molar Carbon Yield of Product of Interest}}{\text{Total Carbon Yield of all Extractable Products}} \times 100\% \qquad (2)$$

The carbon balances of the different reactivity tests can be found in Supplementary Table 1.

## Data availability

All data are available in the main text or the supplementary material. Source data are provided in this paper.

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

## Acknowledgements

This work was financially supported by the IEDO Office of the Department of Energy (DOE), under grant number PS 23A00753. This research used instruments in the Advanced Materials Characterization Lab (AMCL), W. M. Keck Center for Advanced Microscopy & Microanalysis, and the Mass Spectrometry Facility at the University of Delaware. The TGA-MS equipment was supported by the Center for Plastics Innovation, an Energy Frontier Research Center, funded by the U.S. Dept. of Energy, Office of Science, Office of Basic Energy Sciences, under Award Number DE-SC0021166. The authors thank Jennifer Lynch, Katherine Stevens, and other Center for Marine Debris Research members at Hawaii Pacific University for removing and sorting the HDPE fishing nets. The authors also thank Kelly Walker for her assistance with schematics.

## Author contributions

E.S. and K.Y. conceived the idea, designed the project, and conducted the experiments. J.N. carried out catalyst characterization. S.N. performed TGA on the spent catalysts. D.G.V. supervised the project and acquired funding. E.S. and D.G.V. wrote the manuscript, and all authors discussed the results and assisted in manuscript preparation.

## Competing interests

E.S., K.Y., and D.G.V. are inventors on a patent application related to this work filed by the University of Delaware. The remaining authors declare no competing interests.
