## [Peer Review File · Nature Communications]

Recycling polyolefin plastic waste at short contact times via rapid joule heatingREVIEWER COMMENTS

Reviewer #1 (Remarks to the Author):

The manuscript suggests that modern heating techniques e.g., RPH and CJH have been employed to minimize the residence and contact times of polymer chains within the reactor alongside the catalyst. Through rapid pyrolysis and the minimization of secondary reactions, the focus is on maximizing the production of light hydrocarbons while significantly reducing coke formation and aromatics. The manuscript is commendably written and offers a notable novelty. However, it requires closer attention regarding the consistent terminology usage, particularly concerning references to "light olefins" and "light hydrocarbons." Additionally, the method for presenting coke production lacks engagement. It would be more compelling to express coke production relative to the amount of catalyst employed rather than solely in comparison to the polymer quantity, aiming for minimized coke formation. Moreover, the critical parameter of the propylene to ethylene ratio should be addressed within the results section.

Reviewer #2 (Remarks to the Author):

This work investigates the processing variables for rapid joule heating in application of polyolefin conversion to light alkenes (i.e., C₂-C₄ "monomers") over a zeolite catalyst. The work compares to other rapid heating work previously published, albeit without the use of catalysts. The authors vary process heating conditions like pulsing voltage, frequency, duration, and number of pulses as well as catalyst loading and post reaction characterization to determine coke formation. This demonstrates important benchmarking for this reactor design. The manuscript would be attractive for readers, though there are some comments I believe should be addressed before publication.

The authors describe work from Dong et al. that also used a form a Joule heating that achieved "selective monomer production from plastics, even in the absence of a catalyst." However, later they describe it as unselective (lines 130-134). Can the authors clarify? If it is in fact selective without the catalyst for that system, it should be clear in the intro what they hope to improve on.

Do the authors have a sense of the mass balances for this system? Discussion of mass balances often plagues reliable reporting in this area.

Is there any deterioration of carbon fiber over time?

Line 78, CFP is not defined.

As written (line ~97), it seems like the C₂-C₄ are proposed as primary products since it is stated that "short contact times are essential to preclude slower secondary reactions" What does this mean for how the polymer chains are cleaved?

It would be helpful if the selectivity to gas products for the polymer-to-catalyst ratio scans was included in the main text.

It is unclear how many pulses were used in Fig. 2a.

The subheading "effect of pulsing" may be better as "effect of pulse frequency" because the number of pulses was already discussed.

The selectivity component of Fig. 2b is not discussed in the main text.

What is the ratio of the D/G bands in Fig. 4D? They look somewhat similar by eye. It would be helpful to include the same data analysis as the authors use later when discussing Raman.

Line 238, I think this is mean to be "Conclusions" because there is already a results and discussion section.

Has any characterization been done to validate the loss of the aromatic solvents used to disperse the plastic?

Reviewer #3 (Remarks to the Author):

The authors present an interesting paper on the catalytic pyrolysis of polyolefins under pulse heating. Their concept has the potential to provide valuable fundamental insight on chemistry but will arguably be very challenging to adapt and scale-up for continuous commercial application. I therefore strongly recommend profiling this work as fundamental study and refraining any pretention of commercial potential. But then, the study is missing several critical points to provide conclusive fundamental insight. While being carried out carefully, the study misses mass balance, proper benchmark experiments, and necessary experimental details (e.g., on catalyst stability). I therefore recommend major revision that consider the addresses the following shortcomings.

1. Mass balance: the authors are reporting selectivities (likely for gas phase products) that nicely add-up to 100%. But they also report coke deposited on the catalyst, which in one test case accounts for 8 wt% of the feed (Fig. S7A reports ~10 wt% coke on catalysts with Polymer/cat ratio of 1.25). So, I conclude that their selectivity is in fact no true selectivity but a concentration in measurable gaseous product. I therefore strongly recommend developing and reporting proper mass balance, redefining the selectivity accordingly, and highlighting eventual missing products or experimental error.

2. Benchmark experiments: The authors provide a valuable evaluation of process parameters. But I would argue that their comparisons with benchmarks conditions is blurring conclusions by varying multiple variables at the same time.

2a. Since T and reaction time are the classical critical variable, I would recommend maintaining them constant e.g. when varying pulse frequency and length (e.g. Fig. 3). I fear that a large part of the decreasing conversion they observe when decreasing pulse frequency or increasing pulse length is due to a co-variation of T and/or t. As for the time, I would wonder whether the critical time is the 'heating time' of the pulses or the total 'heating + cooling time'. These should be evaluated and disentangled.

2b. They should also run their RPH reactor in continuous mode (at identical T and overall reaction time instead of identical voltage) to provide a fair comparison between pulse and continuous heating using the same reactor configuration and the same hydrodynamics. The comparison with another reactor configuration (e.g. CJH in Fig. 4-5) can then come as additional data to check for other possible effects.

3. Experimental procedure:

3a. The authors should report the overall gas dilution (and thereby product concentration), reports its variation when varying the ratio of heating/non-heating time (or continuous heating) and acknowledge its potential role in evaporating the product and suppressing consecutive reactions. The pulse frequency being much larger than the sampling and measuring time, I indeed expect the analysis to be integral of the several pulses, i.e.

3b. The experimental procedure is not clear for the catalyst stability study. I'm e.g. wondering whether they use the same carbon fiber paper multiple time but loaded it with fresh zeolites. I'm also wondering whether the zeolite was subjected to any regeneration treatment between reuse, e.g. washing or calcination.

3c. I'm wondering how much water was co-fed in the 'steam co-feed' experiments.

Beyond these major points, the authors are also invited to consider a few minor improvements:

4. As the authors use an unusual reaction set up, they should devote a few lines in the text to help the reader understand it, e.g. understand that this is "a flow reactor that consist of a carbon fiber paper impregnated with zeolite and with plastic and is heated by bringing the carbon fiber paper under pulsating potential while He gas is flowing along the paper to remove the vaporized product. The plastic is therefore converted batch wise under stripping gas flow". A simple schematic of the unit would then be valuable, and surely clearer than the pictures provided.

5. The authors should explain all acronyms when using them, e.g. CFP in line 78 or ID/IG in line 200.

6. The authors should avoid the misleading terminology of "upcycling", that is often use as fashionable substitute for recycling. Real "upcycling to much higher value product" is a distraction because the market of high-value product is too small to solve society's waste problem to any significant level. Along similar lines, the authors should avoid the statement of L. 39 that recycling to feedstock would not being circular; it is circular when the feedstock is fed to a steam cracker and the major fraction of its carbon comes out as valuable monomers.

Reviewer #1:

The manuscript suggests that modern heating techniques e.g., RPH and CJH have been employed to minimize the residence and contact times of polymer chains within the reactor alongside the catalyst. Through rapid pyrolysis and the minimization of secondary reactions, the focus is on maximizing the production of light hydrocarbons while significantly reducing coke formation and aromatics. The manuscript is commendably written and offers a notable novelty. However, it requires closer attention regarding the consistent terminology usage, particularly concerning references to "light olefins" and "light hydrocarbons." Additionally, the method for presenting coke production lacks engagement. It would be more compelling to express coke production relative to the amount of catalyst employed rather than solely in comparison to the polymer quantity, aiming for minimized coke formation. Moreover, the critical parameter of the propylene to ethylene ratio should be addressed within the results section.

Authors response

We thank the reviewer for the positive evaluation of our work and the valuable comments. We address all the comments below.

As per the reviewer's suggestions, we have carefully rephrased light olefins and light hydrocarbons as appropriate. Light hydrocarbons are used when we refer to total product distributions, whereas light olefins have been used only at places where we discuss olefin selectivity. We also typically have high selectivity (~90%) to C₂-C₄ olefins.

Previously, we reported the coke produced with respect to polymer-to-catalyst ratios. As per the reviewer's suggestion, we have modified it to a catalyst-to-polymer ratio (Fig. S7), directly indicative of how much coke is produced with increasing/decreasing amounts of catalyst (since the amount of polymer is kept the same throughout the study). We also modified the corresponding discussion in the manuscript and rephrased the following lines.

Fig. S7. (a) Wt% loss of coke on spent catalysts at different catalyst-to-polymer ratios (42 V and 10 pulses) from TGA. (b) An example of TGA weight loss data for spent catalyst obtained from a reaction performed at a catalyst-to-polymer ratio of 0.4.

“At catalyst-to-polymer ratios of <0.4 , heavy coke exceeds 30 wt. % (Fig. S7). Hence, a ratio of ~ 0.8 (polymer-to-catalyst ratio ~ 1.25) was used for subsequent tests.”

We also briefly discussed the P/E ratios and their implications in the conclusions section.

“The high propylene to ethylene ratios (~ 2.6 to 3.5) suggest that the technology is particularly well suited for propylene production.”

Reviewer #2:

This work investigates the processing variables for rapid joule heating in application of polyolefin conversion to light alkenes (i.e., C2-C4 “monomers”) over a zeolite catalyst. The work compares to other rapid heating work previously published, albeit without the use of catalysts. The authors vary process heating conditions like pulsing voltage, frequency, duration, and number of pulses as well as catalyst loading and post reaction characterization to determine coke formation. This demonstrates important benchmarking for this reactor design. The manuscript would be attractive for readers, though there are some comments I believe should be addressed before publication.

We thank the reviewer for our work's positive evaluation and valuable comments and suggestions. We address and clarify all points raised by the reviewer below.

Comment 1: The authors describe work from Dong et al. that also used a form a Joule heating that achieved “selective monomer production from plastics, even in the absence of a catalyst.” However, later they describe it as unselective (lines 130-134). Can the authors clarify? If it is in fact selective without the catalyst for that system, it should be clear in the intro what they hope to improve on.

Authors response

We thank the reviewer for the comment. Indeed, the statement is misleading as it contradicts our previously made statement. Here, we’re reiterating the findings of Dong et al., which demonstrate that the non-catalytic process requires complex phenomena, such as melting, wicking, and reaction, to occur simultaneously to achieve good performance. Without melting and wicking, i.e., with rapid Joule heating alone, depolymerization is unselective. On the contrary, direct Joule heating with a suitable catalyst can selectively depolymerize plastics as long as the pulsing parameters are chosen properly (low frequencies can lead to long cooling periods, which result in lower conversions).

To clarify this point, we have rephrased the following sentence:

“However, with rapid Joule heating alone without a bilayer structure, the non-catalytic depolymerization is unselective due to the lack of a temperature gradient that promotes wicking.”

Comment 2: Do the authors have a sense of the mass balances for this system? Discussion of mass balances often plagues reliable reporting in this area.

Authors response

The carbon balance is >80% at high conversions. At low conversions (<40%), the carbon balance is >90%. This is typical for pyrolysis of plastics because the product stream contains both linear and isomerized hydrocarbons along with a small fraction of cyclic products and aromatics at carbon numbers greater than C5. GC calibrations were performed using linear hydrocarbons and aromatics, and products greater than C₈ carbon number were not quantified (due to very low yields), which explains some of the carbon loss. The carbon balance for different experimental conditions has been included in Table S1.

Comment 3: Is there any deterioration of carbon fiber over time?

Authors response

Based on our reusability tests with both RPH and CJH, we do not observe deterioration of the carbon fiber. Additionally, we have shown in our previous work (Kewei Yu et al., Dynamic Electrification of Dry Reforming of Methane with In Situ Catalyst Regeneration, *ACS Energy Letters*, 2023 8 (2), 1050-1057) that the CFP can be operated continuously for >8 h at temperatures as high as 1000 °C without any deterioration.

Comment 4: Line 78, CFP is not defined.

Authors response

We have defined CFP as carbon fiber paper in line 78 as per reviewer’s suggestion.

“The polymer is in intimate contact with the carbon fiber paper (CFP) and catalyst and is heated resistively very rapidly by the heating element.”

Comment 5: As written (line ~97), it seems like the C2-C4 are proposed as primary products since it is stated that “short contact times are essential to preclude slower secondary reactions” What does this mean for how the polymer chains are cleaved?

Authors response

The polymer chains are cracked as per the well-established monomolecular and bimolecular acid-cracking mechanisms typically involving beta-scission of the tertiary carbenium ions. However, the short contact times minimize the cyclization and aromatization of these olefinic intermediates/products into cyclic and aromatic hydrocarbons. Previous work from Mastalski et al. demonstrates that short times can minimize char formation for thermal pyrolysis (Isaac Mastalski et al., *Chemistry of Materials*, 2023 35 (9), 3628-3639). Likewise, there are multiple pieces of evidence in the catalytic pyrolysis literature that show that short residence times can reduce yields of aromatic products and enhance selectivity towards olefins (Ma del Remedio Hernández et al., *Industrial & Engineering Chemistry Research*, 2006 45 (26), 8770-8778; Son Dong et al., *Applied Catalysis B: Environmental*, 2023, 324, 122219).

We have added the discussion as per the reviewer’s suggestion. The following line was added.

“Furthermore, this finding also corroborates prior literature suggesting that short residence times can suppress the formation of aromatics and enhance selectivity towards olefins for catalytic pyrolysis of polyolefins^{25,26}.”

Comment 6: It would be helpful if the selectivity to gas products for the polymer-to-catalyst ratio scans was included in the main text.

Authors response

They have been included in the main text as per reviewer’s recommendation.

Comment 7: It is unclear how many pulses were used in Fig. 2a.

Authors response

We use 60 pulses for the same and it has been included in figure caption of Fig. 2a, as shown below.

“Figure 2. Performance of thermal and catalytic rapid pulse heating (RPH) on LDPE deconstruction for various operating conditions (a) Effect of DC voltage on the performance of thermal and catalytic RPH (60 pulses).”

Comment 8: The subheading “effect of pulsing” may be better as “effect of pulse frequency” because the number of pulses was already discussed.

Authors response

We have renamed subsection as “*Effect of Pulsing Parameters*”, since we discuss experiments at both varied frequency and constant frequency with varied heating parameters.

Comment 9: The selectivity component of Fig. 2b is not discussed in the main text.

Authors response

We have added the discussion as per the reviewer’s suggestion. The following line was added.

“Additionally, the lower peak and average temperatures result in lower selectivity towards C₂-C₄ hydrocarbons.”

Comment 10: What is the ratio of the D/G bands in Fig. 4D? They look somewhat similar by eye. It would be helpful to include the same data analysis as the authors use later when discussing Raman.

Authors response

We thank the reviewer for the comment. The I_D/I_G ratio for CJH is 0.44 vs. 0.38 for RPH. This again demonstrates greater extent of poly aromatic species in CJH. The following lines have been added to the discussion.

“Raman spectra (Fig. 4D) corroborate significantly higher coking in CJH, demonstrating an I_D/I_G ratio (the ratio of the Defected to Graphitic Carbon) of 0.44 (vs. 0.38 for RPH).”

Comment 11: Line 238, I think this is mean to be “Conclusions” because there is already a results and discussion section.

Authors response

We thank the reviewer for the comment. We have named the section as conclusions.

Comment 12: Has any characterization been done to validate the loss of the aromatic solvents used to disperse the plastic?

Authors response

In order to ensure all residual solvent has been removed, we dry the sample in a vacuum oven once the polymer is coated. We have also performed TGA on the CFP with the polymer deposited on it and do not observe any weight loss corresponding to solvent.

Reviewer #3:

The authors present an interesting paper on the catalytic pyrolysis of polyolefins under pulse heating. Their concept has the potential to provide valuable fundamental insight on chemistry but will arguably be very challenging to adapt and scale-up for continuous commercial application. I therefore strongly recommend profiling this work as fundamental study and refraining any pretention of commercial potential. But then, the study is missing several critical points to provide conclusive fundamental insight. While being carried out carefully, the study misses mass balance, proper benchmark experiments, and necessary experimental details (e.g., on catalyst stability). I therefore recommend major revision that consider the addresses the following shortcomings.

We thank the reviewer for the valuable feedback and comments. We found the comments extremely useful in improving the quality of the manuscript. We have addressed them all to the best of our ability, and our responses are below.

Comment 1. Mass balance: the authors are reporting selectivities (likely for gas phase products) that nicely add-up to 100%. But they also report coke deposited on the catalyst, which in one test case accounts for 8 wt% of the feed (Fig. S7A reports ~10 wt% coke on catalysts with Polymer/cat ratio of 1.25). So, I conclude that their selectivity is in fact no true selectivity but a concentration in measurable gaseous product. I therefore strongly recommend developing

and reporting proper mass balance, redefining the selectivity accordingly, and highlighting eventual missing products or experimental error.

We thank the reviewer for the comment. We report selectivity for non-solid products (i.e., extractable products, meaning gas & liquid products). We see some coking. To avoid confusion, we have changed the term “Selectivity (%)” to “Selectivity of Extractables (%)” in all figures in the manuscript. Concerning the mass balance, the carbon balance is >80% at high conversions. At low conversions (<40%), the carbon balance is >90%. This is typical for pyrolysis of plastics because the products contain both linear and isomerized hydrocarbons along with a small fraction of cyclic products at carbon numbers greater than C₅. GC calibrations were performed using linear hydrocarbons and aromatics, and products greater than C₈ carbon number were not quantified (due to very low yields), which explains some of the carbon loss. We additionally perform some benchmark experiments using a cold trap and analyzed condensed products using a GCMS to ensure we account for all products. Furthermore, we expect improved accuracy if weighing is done using a microbalance, however, the absence of one in our case leads to some errors. The carbon balance for different experimental conditions have been included in Table S1.

Comment 2. Benchmark experiments: The authors provide a valuable evaluation of process parameters. But I would argue that their comparisons with benchmarks conditions is blurring conclusions by varying multiple variables at the same time.

Comment 2a. Since T and reaction time are the classical critical variable, I would recommend maintaining them constant e.g. when varying pulse frequency and length (e.g. Fig. 3). I fear that a large part of the decreasing conversion they observe when decreasing pulse frequency or increasing pulse length is due to a co-variation of T and/or t. As for the time, I would wonder whether the critical time is the ‘heating time’ of the pulses or the total ‘heating + cooling time’. These should be evaluated and disentangled.

We thank the reviewer for the comments. Indeed, we agree that ideally keeping the temperature and time the same and changing just the frequency would be the best measure of this effect. However, the system doesn’t allow temperature control. The temperature is dependent on operational voltage and must be controlled by changing the voltage on the power supply. This is extremely tedious since the power supply can scan between 0-80 V with an incremental span

of 0.1 V. Furthermore, changing frequencies can alter the peak and average temperatures, adding another layer of complexity. Given these challenges, we have tried two approaches to take the reviewer's comments into account. (1) We set the overall power ($V^2/R \cdot t$) to be the same, and keep the total time the same (we do 2 pulses at 0.25 Hz, 4 pulses at 0.5 Hz and 8 pulses at 1 Hz). This can ensure similar average temperatures, but the major issue with this approach is that it can lead to significantly higher peak temperatures ($\sim 900^\circ\text{C}$). Hence, we likely achieve full conversions at all frequencies due to higher peak temps. As expected, we do see similar results in the figure below.

Alternatively, we also tried to maintain the peak temperatures and the total exposure time constant whilst changing frequency (2 pulses at 0.25 Hz, 4 pulses at 0.5 Hz, and 8 pulses at 1 Hz) by altering the voltage. This we believe might be a better comparison for the pulsing frequency and we see that the conversion decreases with decrease in frequency which is mainly

due to lower T_{avg} . The results can be seen in the figure below (Fig. S9) and have been added to the “Effect of pulsing parameters section” in the manuscript”.

The following text was included in the manuscript:

“Additionally, we assessed the effect of pulsing frequency whilst maintaining the peak temperatures ($T_{max} \sim 720$ °C) and the total exposure time (8 seconds) constant. This was accomplished by adjusting the DC voltage and pulse sequences: 2 pulses at 0.25 Hz (65 V), 4 pulses at 0.5 Hz (53V), and 8 pulses at 1 Hz (42 V). Again, the results suggest that the conversions decrease with a decrease in frequency, primarily due to a lower T_{avg} (Fig. S9).”

Comment 2b. They should also run their RPH reactor in continuous mode (at identical T and overall reaction time instead of identical voltage) to provide a fair comparison between pulse and continuous heating using the same reactor configuration and the same hydrodynamics. The

comparison with another reactor configuration (e.g. CJH in Fig. 4-5) can then come as additional data to check for other possible effects.

As explained in the previous comment, identifying the right voltage is challenging. Furthermore, we believe that the comparison provided in the manuscript is fairer since we keep the operational time of the power source constant (10*50 ms in case of RPH vs 0.5 s in case of CJH). This means that the device has energy input for the same amount of time in both cases. However, we have taken into account the reviewer's comment and performed the experiments for 10 s in CJH (10*1 s pulses). At 15 V, $T_{avg} \sim 690\text{ }^{\circ}\text{C}$ – slightly lower than RPH, and 16 V, $T_{avg} \sim 800\text{ }^{\circ}\text{C}$ – slightly higher than RPH. We see that in both cases, we obtain high conversions which is expected since such a long time is not required for the process (even at 0.5 s we reach close to full conversions). Furthermore, TGA analysis of the spent catalysts reveals again that CJH results in with both light and heavy coke being more in CJH than RPH. The results are demonstrated below and included in the SI (Fig. S13).

Fig. S13. (a) Effect of DC voltage on conversion and selectivity for CJH of LDPE over H-ZSM-5 (reaction time = 10 s).

Comment 3. Experimental procedure:

3a. The authors should report the overall gas dilution (and thereby product concentration), reports its variation when varying the ratio of heating/non-heating time (or continuous heating) and acknowledge its potential role in evaporating the product and suppressing consecutive reactions. The pulse frequency being much larger than the sampling and measuring time, I indeed expect the analysis to be integral of the several pulses, i.e.

We thank the reviewer for the comment. In our setup, we do not have an online GC connected to the system - so we don't perform any online sampling. Instead, all products produced were collected in the gas bag throughout the duration of the experiment, followed by injections into a GC-FID. Hence, overall dilution is not a concern, since we control the gas flow rates and measure the exact time for which the products are collected into the gas bag. The total volume of gas collected for each experiment is known. Furthermore, we do not observe any changes in conversion/selectivity with changing gas flow rate suggesting that the carrier gas flow rate doesn't play a role in the reaction and only acts as a carrier gas to remove the products formed. We have added the following lines in the methods section to clarify the same

“The conversion of the plastic wastes was conducted in the JH reactor (Figs. S2-S3) consisting of a CFP impregnated with the zeolite coated with ~10-12 mg of plastic along with an upward He flow of 25 sccm (unless stated otherwise) regulated by a mass flow controller. The He gas flowing along CFP the acts as a carrier gas and removes the vaporized product. A Tedlar bag was connected to the rear end of the reactor section for online collection of the hydrocarbons produced.”

3b. The experimental procedure is not clear for the catalyst stability study. I'm e.g. wondering whether they use the same carbon fiber paper multiple time but loaded it with fresh zeolites. I'm also wondering whether the zeolite was subjected to any regeneration treatment between reuse, e.g. washing or calcination.

We do not perform any washing or regeneration steps while studying catalyst reusability. We use the spent CFP as such after an experiment and coat it with the polymer, following the same procedure for polymer coating described in section. We have highlighted this in the MS in the following lines.

“Prior to each cycle, the CFP was coated with polymer following the procedure described in the Methods section without washing or regenerating the catalyst.”

3c. I'm wondering how much water was co-fed in the 'steam co-feed' experiments.

In this work, we bubble the He gas through a bubbler containing water at room temperature for the steam co-feeding experiments. Hence, the concentration of water depends on the vapour pressure of water and is calculated to be ~3-4 %.

Beyond these major points, the authors are also invited to consider a few minor improvements:

Comment 4. As the authors use an unusual reaction set up, they should devote a few lines in the text to help the reader understand it, e.g. understand that this is “a flow reactor that consist of a carbon fiber paper impregnated with zeolite and with plastic and is heated by bringing the carbon fiber paper under pulsating potential while He gas is flowing along the paper to remove the vaporized product. The plastic is therefore converted batch wise under stripping gas flow”. A simple schematic of the unit would then be valuable, and surely clearer than the pictures provided.

We thank the reviewer for the comment. We have included a more detailed schematic in the SI. We have also added a few lines in the methods section under the reactivity tests sub-section describing this.

Fig. S1. Schematic of experimental setup for Joule heating reactor.

Fig. S3. (a) Experimental setup for Joule heating reactor. (b) CFP - heating element. (c-d) Zoomed-in snapshot of heating element inside the reactor and position of the IR camera. (e) Sample temperature profile of CFP obtained using an IR camera.

“The conversion of the plastic wastes was conducted in the JH reactor (Figs. S2-S3) consisting of a CFP impregnated with the zeolite coated with ~10-12 mg of plastic along with an upward He flow of 25 sccm (unless stated otherwise) regulated by a mass flow controller. The He gas flowing along CFP the acts as a carrier gas and removes the vaporized product. A Tedlar bag

was connected to the rear end of the reactor section for online collection of the hydrocarbons produced.”

Comment 5. The authors should explain all acronyms when using them, e.g. CFP in line 78 or ID/IG in line 200.

We thank the reviewer for the comment. We have carefully abbreviated both acronyms.

“The polymer is in intimate contact with the carbon fiber paper (CFP) and catalyst and is heated resistively very rapidly by the heating element.”

“Raman spectra (Fig. 4D) corroborate significantly higher coking in CJH, demonstrating an ID/IG ratio (the ratio of the Defected to Graphitic Carbon) of 0.44 (vs. 0.38 for RPH).”

Comment 6. The authors should avoid the misleading terminology of “upcycling”, that is often used as fashionable substitute for recycling. Real “upcycling to much higher value product” is a distraction because the market of high-value product is too small to solve society’s waste problem to any significant level. Along similar lines, the authors should avoid the statement of L. 39 that recycling to feedstock would not be circular; it is circular when the feedstock is fed to a steam cracker and the major fraction of its carbon comes out as valuable monomers.

We thank the reviewer for the comment. We only use the term “upcycling” when the products produced are other than monomers. For example, hydroconversion can produce lubricants or alkyl aromatics, which are estimated to have higher MSPs than monomers. We have also rephrased sentences regarding circularity of processes as per the reviewer’s suggestion. The following lines have been rephrased in the introduction.

“Consequently, there has been a surge in chemical upcycling and recycling technologies.”

“Nevertheless, these processes are not directly circular, since their products need additional processing to produce monomers.”

REVIEWER COMMENTS

Reviewer #1 (Remarks to the Author):

The manuscript has undergone another reading, and it appears that all identified flaws have been addressed, rendering it prepared for publishing.

Reviewer #2 (Remarks to the Author):

The authors satisfactorily addressed my comments as well those of other reviewers. Therefore, I think it is suitable to publish. One minor comment: I looked at this version on a different computer, and it is very hard to see the difference between C5-C7 and C8+ colors shown in the bar graphs, regardless of the gradient color.

Reviewer #3 (Remarks to the Author):

I thank the authors for accommodating most of my comments. I still feel that some need to be addressed better.

I maintain my comment that the experiments reported have limited technology potential, they are mainly valuable for fundamental understanding. They present new lab setup to explore chemistry but no "technology" or "process". So, I strongly recommend moderating the wording and promises proposed in the introduction and conclusions, e.g.

"Here, 17 we demonstrate a single-step electrified process utilizing Rapid Joule Heating (RPH) over H 18 ZSM-5 catalyst to efficiently deconstruct polyolefin plastic ..."

"The proposed Rapid Pulse Joule Heating (RPH) technology for catalytic cracking of polyolefin waste exhibits significant promise in achieving high monomer selectivity."

I invite the authors to be more thorough in their upcycling promises, e.g. with "Chemical upcycling of polyolefins to fuels, lubricants, and waxes offers a promising strategy for mitigating their accumulation in landfills and the environment." This remains misleading. Conversion to fuel helps the mitigating the accumulation of waste but is downcycling; conversion to wax is upcycling but too small a scale to mitigate waste accumulation.

Comment 1.

I maintain not supporting the definition of selectivity used by the authors and defined in line 367. What they call selectivity is in fact fraction of gas+liquid products. A true selectivity is defined as yield/conversion and is lowered by the co-production of coke (on catalyst & on supporting paper) and by (inevitable) loss of products, both of which are present here. Moreover, the selectivity defined in line 367 should be converted to C% as mole% are providing a very strong bias towards low C_n products. I invite the authors to correct that, at a minimum by renaming selectivity as "fraction of gas+liq product" and expressed it in C%. Much better, though, would be to provide convert their 'selectivities' into 'yield' (by multiplying their 'selectivity' by their 'conversion') or maintain them as selectivities but correcting for loss coke and C-losses.

I also challenge their definition of conversion (line 359) which is truly a degree of volatilization. The true conversion could be higher than reported when the polymer is converted to coke that remains on the catalyst of CFP. Such comment should be added to the paper, e.g. in "product analysis".

They should also mention C balances of 80-98% in this section and refer to table S1 of SI for more details. They authors should also check for possible inversion of T_{max} and T_{avg} for the bottom 2 experiments.

Comment 2.

I thank the authors for explaining the technical challenges in controlling the truly relevant process parameters and running additional control experiments. In view of this, the authors should provide more information in the main text. For instance, they should report the T_{max} for every stack bar of every figure to help the reader better understand the observations. The authors should then try to explain their observations (e.g. on pulsing frequency) in terms of direct parameters T_{max} and total heating time, rather than indirect ones such as number and frequency of pulses.

NB: The T_{avg} is likely less critical if it is indeed the average of T_{max} (up to 1000 C) and T_{min} (~450 C) and the reaction rate is following the Arrhenius law.

Comment 3.

Thanks for clarifying the experimental setup. As a reader, it would help me to see fig. S2 and read the short paragraph BEFORE reading the results, and not after. Can the authors move this information to a better place? Could they also mention the water content of 3-4% in the steam co-feed experiments?

Reviewer #1:

The manuscript has undergone another reading, and it appears that all identified flaws have been addressed, rendering it prepared for publishing.

Authors response

We thank the reviewer for the positive evaluation of our work.

Reviewer #2:

The authors satisfactorily addressed my comments as well those of other reviewers. Therefore, I think it is suitable to publish. One minor comment: I looked at this version on a different computer, and it is very hard to see the difference between C5-C7 and C8+ colors shown in the bar graphs, regardless of the gradient color.

Authors response

We thank the reviewer for the comment and our work's positive evaluation. As per reviewer's suggestion we have changed the colors of the C8+ products in the all the graphs to enhance readability. Shown below is the updated version of Figure 2 as an example.

Reviewer #3:

I thank the authors for accommodating most of my comments. I still feel that some need to be addressed better.

I maintain my comment that the experiments reported have limited technology potential, they are mainly valuable for fundamental understanding. They present new lab setup to explore chemistry but no “technology” or “process”. So, I strongly recommend moderating the wording and promises proposed in the introduction and conclusions, e.g.

“Here, 17 we demonstrate a single-step electrified process utilizing Rapid Joule Heating (RPH) over H 18 ZSM-5 catalyst to efficiently deconstruct polyolefin plastic ...”

“The proposed Rapid Pulse Joule Heating (RPH) technology for catalytic cracking of polyolefin waste exhibits significant promise in achieving high monomer selectivity.”

I invite the authors to be more thorough in their upcycling promises, e.g. with “Chemical upcycling of polyolefins to fuels, lubricants, and waxes offers a promising strategy for mitigating their accumulation in landfills and the environment.” This remains misleading. Conversion to fuel helps the mitigating the accumulation of waste but is downcycling; conversion to wax is upcycling but too small a scale to mitigate waste accumulation.

Authors response

We thank the reviewer for the valuable feedback and comments. We have addressed them all to the best of our ability, and our responses are below. We have also rephrased the sentences highlighted above and other sentences throughout the manuscript to moderate the claims in the introduction and conclusions as per the reviewer’s suggestions. We have rephrased all discussions regarding selectivity to product fraction and added some discussion regarding limitations of our approach the end of the discussion section. Listed below are the rephrased sentences.

Abstract

“The chemical deconstruction of polyolefins to fuels, lubricants, and waxes offers a promising strategy for mitigating their accumulation in landfills and the environment.”

“Here, we demonstrate a single-step electrified approach utilizing Rapid Joule Heating (RPH) over H-ZSM-5 catalyst to efficiently deconstruct polyolefin plastic waste into light olefins (C₂-C₄) in milliseconds with high productivity at much higher polymer to catalyst ratio than prior work.”

“The catalyst is essential in producing a narrow distribution of light olefins. Pulsed operation and steam co-feeding enable highly selective deconstruction (product fraction of >90% towards C₂-C₄ hydrocarbons) with minimal catalyst deactivation compared to Continuous Joule Heating (CJH).”

“This approach demonstrates effective deconstruction of various real-life waste materials, resilience to additives and impurities, and versatility for circular polyolefin plastic waste management.”

Introduction

“This work presents a laboratory-scale approach to selectively deconstruct polyolefins into C₂-C₄ olefins using Rapid Pulse Joule Heating (RPH) over an H-ZSM-5 catalyst (Pathway 2 in Fig. 1).”

“We investigate the influence of different reactor parameters on polymer conversion and product distributions.”

“Additionally, we demonstrate that co-feeding steam enhances the yield of light olefins and reduces coking.”

Results and Discussions

“Reaction conditions were optimized to maximize conversion and product fractions of monomers”

“However, the product distributions differ profoundly: non-catalytic cracking leads to a broad distribution of alkanes and olefins ranging from C₁-C₃₁ along with heavier wax products (Fig. S4-5), whereas catalytic cracking results in complete conversion and produces C₁-C₈ range hydrocarbons, with a product fraction of ~74% for C₂-C₄ products. A voltage of 42 V achieved

the highest product fraction of C₂-C₃ products (~45%) and was selected for subsequent tests. We attribute the high product fractions of light olefins to the thin polymer film (<20 μm thick) and short contact times.”

“Additionally, the lower peak and average temperatures results in lower fractions of C₂-C₄ hydrocarbons.”

“10 pulses gave no distinct change in conversion and product fractions, likely due to a similar T_{max} (~690 – 750 °C)”

“Similar trends were seen for product fractions, with an increase in heavier products at lower temperatures.”

“The fraction of C₂-C₃ products also increased with voltage, plateauing around 26-27 V (Fig. 4A). Further increasing the voltage enhanced the yields of lighter products, including methane, acetylene, and propyne. Our results demonstrated that CJH exhibits slightly higher product fractions of C₂-C₄ products compared to RPH”

“While CJH slightly increases the fraction of light olefins produced compared to RPH, pulsed operation often offers other co-benefits, for instance, it can reduce coke due to shorter exposure at elevated temperatures”

“Fig. 4B illustrates that, as hypothesized, the LDPE conversion and product distributions remain unchanged in 3 cycles of reuse in RPH. In contrast, CJH exhibits a decline in conversions and fraction of light olefins produced upon reuse”

“The results (Fig. 5A) demonstrate a substantial enhancement in the fractions of C₂-C₄ olefins when steam was co-fed in RPH”

“Similarly, we also observe a slight increase in the C₂-C₄ olefin fractions when steam was co-fed in CJH.”

“Fig. 6A illustrates that the conversions and product distributions remain largely unaffected, irrespective of source, highlighting the process’s resilience. Notably, PP materials yield a higher fraction of C₂-C₄ range products”

Discussion

“The proposed Rapid Pulse Joule Heating (RPH) for catalytic cracking of polyolefin waste exhibits significant promise in depolymerization of polyolefins to monomers.”

“H-ZSM-5 catalyst enables the effective breakdown of polymers into light hydrocarbons, with a C₂-C₄ product fraction of >75% at full conversion.”

“Notably, we demonstrate a product fraction of >90% towards C₂-C₄ hydrocarbons at full conversion, highlighting the RPH's potential for monomer production.”

“Furthermore, RPH's efficacy with various real-life feedstocks opens new avenues for monomer production from polyolefins, presenting a sustainable approach while mitigating associated CO₂ emissions.”

Limitations of our approach

“While the proposed electrified reactor demonstrates promise for monomer production from plastics waste, it is only a prototypical laboratory-scale framework, which needs to be scaled up for commercialization.”

Comment 1. I maintain not supporting the definition of selectivity used by the authors and defined in line 367. What they call selectivity is in fact fraction of gas+liquid products. A true selectivity is defined as yield/conversion and is lowered by the co-production of coke (on catalyst & on supporting paper) and by (inevitable) loss of products, both of which are present here. Moreover, the selectivity defined in line 367 should be converted to C% as mole% are providing a very strong bias towards low C_n products. I invite the authors to correct that, at a minimum by renaming selectivity as “fraction of gas+liq product” and expressed it in C%. Much better, though, would be to provide convert their ‘selectivities’ into ‘yield’ (by multiplying their ‘selectivity’ by their ‘conversion’) or maintain them as selectivities but correcting for loss coke and C-losses.

I also challenge their definition of conversion (line 359) which is truly a degree of volatilization. The true conversion could be higher than reported when the polymer is converted to coke that remains on the catalyst of CFP. Such comment should be added to the paper, e.g. in “product analysis”.

They should also mention C balances of 80-98% in this section and refer to table S1 of SI for more details. They authors should also check for possible inversion of T_{max} and T_{avg} for the bottom 2 experiments.

Authors response

We thank the reviewer for the comment. We agree that it would be best to report selectivity as selectivity corrected for coke and other carbonaceous losses. However, for our system this is an extremely difficult and tedious process as it involves separating the catalysts from the CFP and performing TGA on every sample tested. Furthermore, this procedure cannot be applied for the reusability tests as removing the catalyst after each cycle is not feasible. Hence, taking these limitations into account we have redefined the term “Selectivity of Extractables (%)” as “Product fraction of Extractables (%)” as per the reviewer’s suggestion. We would also like to clarify that the selectivities we have been reporting thus far are on a Carbon mole (%) basis, and we apologize for the lack of clarity. The methods section has modified the following lines to clarify the same.

“The product fraction of extractable products (gas and liquids) was calculated as follows:

$$\text{Product fraction of Extractables (\%)} = \frac{\text{Molar Carbon Yield of Product of Interest}}{\text{Total Carbon Yield of all Extractable Products}} \times 100 \%$$

Furthermore, as per the reviewer’s recommendation, we have added a statement that the true conversion of the polymer is likely to be higher than the reported conversion since we consider the polymer into coke as an unconverted polymer as per our definition of conversion. The following lines have been added to the methods section to clarify.

“It must be noted that the calculated conversion here does not consider the inevitable loss of polymer into coke (typically <10% based on TGA). Hence, the real conversion is expected to be slightly higher than the reported conversion.”

We apologize for the error and thank the reviewer for catching this error. Indeed, the temperatures in the table for the bottom to entrees in Table S1 were inverted and have been corrected now. We have also included a statement about the carbon balances in the main text.

Comment 2. I thank the authors for explaining the technical challenges in controlling the truly relevant process parameters and running additional control experiments. In view of this, the authors should provide more information in the main text. For instance, they should report the T_{max} for every stack bars of every figures to help the reader better understanding the observations. The authors should then try to explain their observations (e.g. on pulsing frequency) in terms of direct parameters T_{max} and total heating time, rather than indirect ones such as number and frequency of pulses.

NB: The T_{avg} is likely less critical if it is indeed the average of T_{max} (up to 1000 C) and T_{min} (~450 C) and the reaction rate is following the Arrhenius law.

Authors response

We thank the reviewer for the comments. We have presented the T_{max} and T_{avg} values for all the conditions used throughout the manuscript in Table S1. Additionally, in places where we vary the reaction/pulsing conditions, we have included the T_{max} in the figure, figure caption, or the main text, depending on how space permits. We have included the T_{max} or T_{avg} in parenthesis wherever we discuss these effects throughout the results section. Indeed, we agree with the reviewer that T_{max} is the more dominant factor and observe that T_{avg} mainly influences deactivation rather than reactivity. Shown below are examples of figures that have T_{max} included. We have also highlighted the areas where we discuss temperature effects in the resubmitted manuscript.

Comment 3. Thanks for clarifying the experimental setup. As reader, it would help me to see fig. S2 and read the short paragraph BEFORE reading the results, and not after. Can the authors move this information to better place? Could they also mention the water content of 3-4% in the steam co-feed experiments? reactions.

We thank the reviewer for the comment. As recommended by the reviewer, we have added a brief discussion of the reactor setup referencing Fig. S2 at the beginning of the results section for a better understanding of the experimental setup.

The following lines have been included:

“The depolymerization of the plastic wastes was conducted in a Joule Heated reactor (Fig. S2-S3) consisting of a carbon fiber paper (CFP) impregnated with an MFI zeolite (H-ZSM-5) and coated with the plastic. The polymer is in intimate contact with the CFP and the catalyst and is heated resistively very rapidly by the heating element. The flowing He gas, regulated by a

mass flow controller, entrains the gas products and minimizes their contact with the catalyst and secondary reactions.”

We have also added details regarding steam co-feeding in the methods section of the main text.

“For experiments involving steam co-feeding, the He gas was bubbled through a bubbler containing water, resulting in a concentration of ~3-4% water in the gas stream.”

REVIEWERS' COMMENTS

Reviewer #3 (Remarks to the Author):

I thank the authors for addressed quite a few of requests for clarification or corrections.

I feel they have not properly addressed some remaining ones, however:

I value the clarification brought by specifying T_{max} with the results. However, I still find regrettable that the authors relate their results to equipment variables rather than true chemistry variables. They indeed report the product fraction as function of voltage, pulse frequency, heating time per pulse, etc. rather than T_{max} , time at T_{max} , or any suitable severity factor to be defined. The chemistry variables are the ones that could provide insight and could be translated to other equipment. The equipment variables are limited to very similar equipment and can't be transposed to other types, design or size.

More importantly, the limitations of the approach are much more severe than reporting a "prototypical laboratory-scale framework, which needs to be scaled up for commercialization", as state by the authors. The prototype proposed is incompatible with industrial scale:

- A full-scale process can't affordably operate in batch mode with plastic waste being sandwiched between heaters,
- It also needs to reproduce the profiles in time, temperature and likely also gas dilution along the journey of the plastic waste through the reactor to reproduce the yields results reported, and
- The recovery and purification of the olefins, which is presently done by cryogenic distillation, will most likely be exorbitant with the gas dilution reported here and deemed instrumental for reaching the yields reported.

Hence, I strongly recommend the author to present their results (in the abstract, discussion and conclusions) as academic results that have no pretention of commercial application.

Reviewer #3:

I thank the authors for addressed quite a few of requests for clarification or corrections.

I feel they have not properly addressed some remaining ones, however:

I value the clarification brought by specifying T_{max} with the results. However, I still find regrettable that the authors relate their results to equipment variables rather than true chemistry variables. They indeed report the product fraction as function of voltage, pulse frequency, heating time per pulse, etc. rather than T_{max} , time at T_{max} , or any suitable severity factor to be defined. The chemistry variables are the ones that could provide insight and could be translated to other equipment. The equipment variables are limited to very similar equipment and can't be transposed to other types, design or size.

More importantly, the limitations of the approach are much more severe than reporting a "prototypical laboratory-scale framework, which needs to be scaled up for commercialization", as state by the authors. The prototype proposed is incompatible with industrial scale:

- A full-scale process can't affordably operate in batch mode with plastic waste being sandwiched between heaters,
- It also needs to reproduce the profiles in time, temperature and likely also gas dilution along the journey of the plastic waste through the reactor to reproduce the yields results reported, and
- The recovery and purification of the olefins, which is presently done by cryogenic distillation, will most likely be exorbitant with the gas dilution reported here and deemed instrumental for reaching the yields reported.

Hence, I strongly recommend the author to present their results (in the abstract, discussion and conclusions) as academic results that have no pretention of commercial application..

Authors response

We thank the reviewer for their valuable comments. We have taken into account the reviewers comments and highlighted the limitations of our work with respect to scale-up and commercialization in the manuscript. We have modified/added the following statements in the Abstract and Discussion sections:

Abstract

“This laboratory-scale approach demonstrates effective deconstruction of real-life waste materials, resilience to additives and impurities, and versatility for circular polyolefin plastic waste management.”

Discussion

“While the proposed electrified reactor demonstrates promise for monomer production from plastic waste, it is currently a prototypical laboratory-scale framework with limited potential for commercialization. Further development is necessary, particularly in reactor design to facilitate scale-up and continuous operation, as well as in optimizing product recovery methods. Additionally, benchmarking at larger scales is essential to facilitate practical large-scale implementation.”